# Deep learning for EEG-based Motor Imagery classification: Accuracy-cost trade-off

Javier León[1]*, Juan José Escobar[1], Andrés Ortiz[2], Julio Ortega[1], Jesús González[1], Pedro Martín-Smith[1], John Q. Gan[3], Miguel Damas[1]

**1** Department of Computer Architecture and Technology, University of Granada, Granada, Spain, **2** Department of Communications Engineering, University of Málaga, Málaga, Spain, **3** School of Computer Science and Electronic Engineering, University of Essex, Colchester, United Kingdom

* jaleon@correo.ugr.es

## Abstract

Electroencephalography (EEG) datasets are often small and high dimensional, owing to cumbersome recording processes. In these conditions, powerful machine learning techniques are essential to deal with the large amount of information and overcome the curse of dimensionality. Artificial Neural Networks (ANNs) have achieved promising performance in EEG-based Brain-Computer Interface (BCI) applications, but they involve computationally intensive training algorithms and hyperparameter optimization methods. Thus, an awareness of the quality-cost trade-off, although usually overlooked, is highly beneficial. In this paper, we apply a hyperparameter optimization procedure based on Genetic Algorithms to Convolutional Neural Networks (CNNs), Feed-Forward Neural Networks (FFNNs), and Recurrent Neural Networks (RNNs), all of them purposely shallow. We compare their relative quality and energy-time cost, but we also analyze the variability in the structural complexity of networks of the same type with similar accuracies. The experimental results show that the optimization procedure improves accuracy in all models, and that CNN models with only one hidden convolutional layer can equal or slightly outperform a 6-layer Deep Belief Network. FFNN and RNN were not able to reach the same quality, although the cost was significantly lower. The results also highlight the fact that size within the same type of network is not necessarily correlated with accuracy, as smaller models can and do match, or even surpass, bigger ones in performance. In this regard, overfitting is likely a contributing factor since deep learning approaches struggle with limited training examples.

## 1 Introduction

Over the last decades, computational power has experienced significant increases thanks to a wide array of new technologies and computing paradigms. As a result, many problems can now be tackled and many lines of research have appeared with the advances in other fields. In particular, bioinformatics attempts to understand biological data through computer science, mathematics, and statistics. Applications to gene expression analysis [1–3] or brain activity analysis are popular examples.

**Data Availability Statement:** All relevant datasets are available in the paper and its Supporting Information files, and under the following URL:

(https://atcproyectos.ugr.es/hpeecobe/index.php?menu=deliverables).

**Funding:** Javier León, Julio Ortega, Jesús González, Miguel Damas, Pedro Martín-Smith, Juan José Escobar Pérez: Grant number PGC2018-098813-B-C31 (Spanish Ministerio de Ciencia, Innovación y Universidades, http://www.ciencia.gob.es/portal/site/MICINN/) Andrés Ortiz: Grant numbers PGC2018-098813-B-C32 and PSI2015-65848-R (Spanish Ministerio de Ciencia, Innovación y Universidades, http://www.ciencia.gob.es/portal/site/MICINN/) The funders had no role in study design, data collection and analysis, decision to publish, or preparation of the manuscript.

**Competing interests:** The authors have declared that no competing interests exist.

Brain activity can be recorded and analyzed in several ways through Brain-Computer Interfaces (BCIs). BCI paradigms can be divided into three categories: invasive, partially invasive, and non-invasive. Invasive procedures like visual [4, 5] or motor [6] implants are the most powerful, but they also carry many risks derived from surgery such as scar tissue, infections, or rejection. On the other hand, non-invasive procedures such as Electroencephalography (EEG) or Functional Magnetic Resonance Imaging (fMRI) can aid medical diagnosis [7] and research [8], and also tackle real-world problems [9]. Electrocorticography (ECoG) [10, 11] represents a partially invasive middle ground, where surgery is still needed but only to place devices on the surface of the brain.

EEG analysis is the focus of this paper. To record brain activity, slight voltage changes in the brain are measured with a set of electrodes placed on the scalp. Although as a non-invasive method it has poor spatial resolution (it is hard to locate the area that originated the activity), its high sampling rate provides high temporal resolution. Flexibility and ease of use are its main advantages over the other alternatives. However, it still presents some challenges that can be faced with the help of machine learning. EEG, like many other biological sources of data, is known for producing samples with a high dimensionality, i.e., a large number of features. Moreover, observations are in turn scarce, often due to the cost of data acquisition. In this case, for each recording session, subjects are often asked to perform certain actions in order to obtain new data. Repeating this process many times involves a considerable inconvenience for both researchers and participants.

Derived from this type of data, the curse of dimensionality problem [12] is usually present, as the features vastly outnumber the observations. In the particular case of this work, machine learning algorithms may lose ability to generalize knowledge. A possible solution is Feature Selection (FS), which brings several benefits: noise and redundancy removal, reduced computational costs, and improved classification accuracy.

FS is often highlighted in the existing literature on BCI applications [13], citing its importance in real-time performance or the understanding of the brain, among other benefits. However, FS is an NP-hard problem [14], which renders brute-force approaches unfeasible due to the size of the search space. The three main types of alternative methods are filter, wrapper and embedded [15]. Filter methods measure the relationship between features and the dependent class variable. Wrapper methods evaluate the performance of a classifier using different feature subsets. Embedded methods integrate FS into the classifier. The advantage of filters lies in their lower computational complexity, whereas wrapper and embedded approaches frequently achieve better results. In this paper, a wrapper method based on a Genetic Algorithm (GA) is employed. GAs are popular for BCI tasks, be it for FS [16–18] or other purposes [19].

Neural networks are a promising alternative to address the complexity of BCI data, since they are universal approximators and thus they can represent a wide variety of continuous functions. Besides standard Feed-Forward Neural Networks (FFNNs), which are the simplest kind of neural network in terms of structural design, there is growing interest in architectures that are able to leverage context. Convolutional Neural Networks (CNNs) extract local patterns through the convolution operator, and have been successfully applied to EEG signals [20, 21]. Recurrent Neural Networks (RNNs) [22, 23], which are not as widespread as CNNs yet, can dynamically store context to improve processing of individual bits of data.

In EEG-based Motor Imagery (MI), many machine learning algorithms and feature extraction methods have been studied to try to overcome the limitations of small dataset and poor signal-to-noise ratios. The Support Vector Machine (SVM), despite its age, can still produce promising results when paired with the right features: in [24], mutual information is calculated from Common Spatial Pattern (CSP) features to select optimal frequency bands, and dimensionality is further reduced by means of Linear Discriminant Analysis (LDA) before

finally classifying the patterns with SVM; in a later study [25], the same authors use LDA for spatial filtering and a Long Short-Term Memory (LSTM) network for temporal filtering before classifying again with SVM. Alternative approaches to classic machine learning also exist, such as classification by Riemannian geometry [26] or by a residual norm-based strategy [27].

Deep learning-based proposals have also become fairly common in the last few years. A 4-layer FFNN is the model of choice in [28], where the authors successfully combine EEG signals with readings of hemodynamic responses to increase classification accuracy. LSTM, which is a type of RNN, is used this time as a classifier in [23]. In the study, the EEG signals are preprocessed to favor context extraction, which is what RNNs excel at. In [29], four different CNNs with increasing depths are used to learn temporal and spatial features that are then fused and fed to either a multi-layer perceptron or an autoencoder for classification. CNNs are also explored in [30] in combination with FBCSP, a spatial filtering algorithm, and their own approach to extract temporal features. In [31], a recurrent convolutional network (called RCNN) is implemented to also leverage the ability of RNNs to store context, aside from a standard CNN model (called pCNN). The authors compare their proposals to two other CNNs from the state-of-the-art, a shallow one and a deep one. The deep one is found to be as competitive as their pCNN, but they make the case for pCNN by conducting a real-time experiment where the deep CNN has a longer delay of roughly 2.5 seconds against 1.4 of pCNN. Indeed, the feasibility of less complex networks can be worth exploring with the goals of real-time responsiveness and cost-saving in mind. The trade-off between quality and its associated cost is not usually given much relevance, as the search for reliable classification frameworks across sessions and test subjects is already a daunting task.

In this paper, we analyze the performance of three types of neural networks (CNN, FFNN, and RNN) after a GA-based hyperparameter optimization procedure. We compare the three architectures among themselves and to previous work in terms of classification accuracy, but we also place emphasis on energy and time consumption during training, which need to be taken into account when creating a BCI framework but are often relegated to the background. Moreover, since the experiments are repeated several times, we take a look at the variability of network sizes within the same architecture to determine the relationship between complexity and accuracy. In our previous work [32], we optimized and compared CNNs and SVMs. The optimization process for CNN in [32] dealt only with learning hyperparameters, which is now the second step of the method presented here. SVMs are not considered again, since the results in [32] showed a substantial quality loss with respect to CNNs.

This paper is organized as follows: Section 2 describes the data; Section 3 explains the neural network models, the genetic algorithm, the performance measurements, and the statistical analysis; Section 4 contains the experimental results; finally, the conclusions can be found in Section 5.

## 2 The datasets

The EEG datasets used to evaluate the proposed procedures were recorded in the BCI Laboratory at the University of Essex, UK, specifically collected for the purpose of the study reported in [33]. This paper presents a continuous study aiming to improve the results of the above paper. There were 12 healthy subjects recruited for the BCI experiment, aged from 24 to 50 (58% female, 50% naïve to BCI). Before the experiment, the subjects gave their written informed consent using a form approved by the Ethics Committee of the University of Essex and were paid for their participation. More details about this dataset can be found in [33].

The EEG data used here was generated by MI. In this paradigm, limb movement imagination produces a series of brief amplifications and attenuations: Event-Related

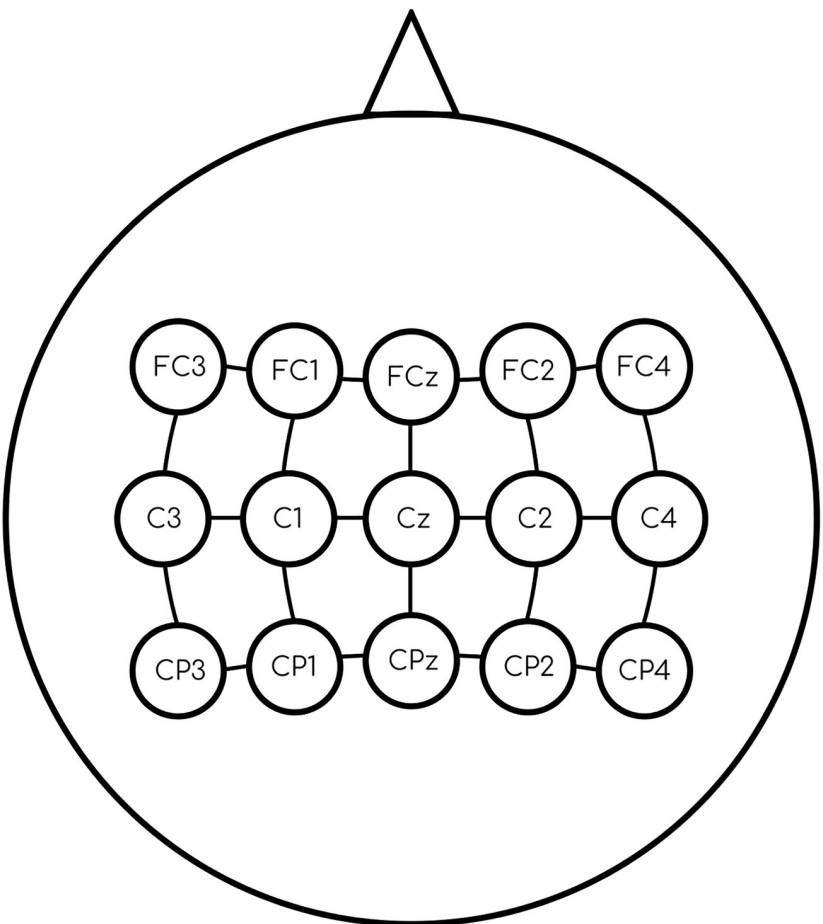

**Fig 1. Electrodes used in the recording process.** Placement of the 15 electrodes according to the extension of the International 10-20 system.

Desynchronization (ERD) and Event-Related Synchronization (ERS), respectively. EEG trials were recorded from 15 electrodes (see Fig 1) at a sampling rate of 256 Hz, and were used to create patterns through the discrete wavelet transform, a type of Multiresolution Analysis (MRA) [34], as seen in [33].

The signal obtained from each electrode was divided into consecutive and partially overlapping segments (20 segments in total). MRA was performed for each segment with 6 wavelet levels to produce sets of coefficients of two types, approximation and details, with decreasing size in powers of two (128, 64, 32, 16, 8, and 4 coefficients for levels 6 to 1). Altogether, an EEG pattern has $2 \times S \times E \times L$ sets of coefficients, where $S$ is the number of segments, $E$ is the number of electrodes, and $L$ is the number of wavelet levels. With the parameters already described, $S = 20$, $E = 15$, and $L = 6$. This means 3, 600 sets of coefficients, and a total of 151, 200 coefficients. By computing the within-set variance, that number can be reduced to 3, 600 coefficients [33]. The resulting patterns are also normalized between 0 and 1, and no missing values are present in the dataset.

The amount of patterns for training and testing is 178, with each pattern containing 3, 600 features. Since the ratio between sample size and number of features is still far from ideal, the

classification could benefit from a further dimensionality reduction. In this regard, wrapper and filter multi-objective evolutionary FS techniques were proposed in [16, 35].

Thus, the aim of this paper is the classification of EEG patterns into three classes that represent imagined left and right hand movements and imagined feet movement. Due to time constraints, three datasets with balanced classes, corresponding to the most promising three BCI subjects (104, 107, and 110), will be used for the experiments. In addition to model accuracy, emphasis is also placed on computational efficiency by measuring time and energy consumption. More in-depth studies on the smart use of available computing devices applied to our datasets can be found in [36, 37].

## 3 Methodology

In this section, the three types of neural networks considered in this paper are presented, along with the genetic algorithm used in the optimization procedure. Afterwards, the quality measures and the statistical analysis are also described.

### 3.1 Feed-Forward Neural Networks

The Feed-Forward Neural Network (FFNN) [38] is the simplest kind of artificial neural network. Loosely based on the human brain, it is composed of processing units called neurons that are organized in three kinds of layers: input, output, and hidden layers. The input layer contains as many units as the number of features in the data. The output layer has one neuron for each possible data class (classification) or a single neuron (regression). The hidden layers are inserted between the input and output layer.

The non-linear boundary approximation capabilities of the network are given by the connections between neurons of adjacent layers. As the name implies, connections in FFNNs are strictly one-way starting from the input layer, and the input values are successively transformed by hidden layers until they reach the output layer (see Fig 2).

FFNNs are usually fully-connected. This means that, in principle, every neuron from a given layer passes its output to every neuron of the next layer. The weights of the connections, learned when the network is trained, dictate exactly how these interactions happen. For instance, for the structure depicted in Fig 2, the total input of a certain neuron $j$ in $L_2$ (the

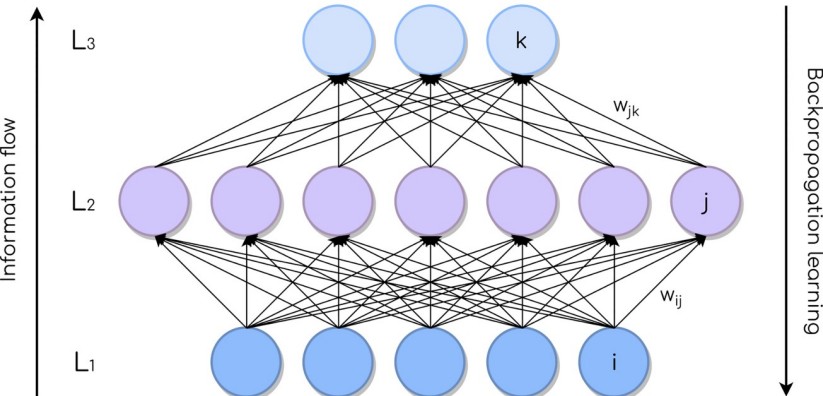

**Fig 2. FFNN with one layer of each kind: Input, hidden, and output (bottom to top).** Connections between neurons are depicted with arrows, and every one of them has an associated weight.

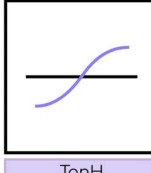 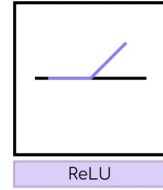 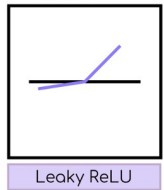 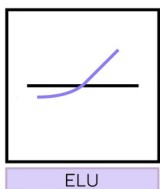

| TanH | ReLU | Leaky ReLU | ELU |

**Fig 3. TanH, ReLU, Leaky ReLU, and ELU activation functions.**

hidden layer) can be written as in Eq 1:

$$z_j = \sum_{i \in L_1} w_{ij} \cdot y_i, \tag{1}$$

where $y_i$ is the output of the $i$-th neuron in $L_1$. The output of a neuron is determined by a nonlinear function of its total input called the *activation function*. There are a number of activation functions in the literature, of which some of the most common are (see also Fig 3):

- *Hyperbolic tangent (TanH)*: a widely used function due to being bounded, which increases training efficiency. However, its shape at both ends can be problematic for gradient computation in backpropagation.

- *Rectifier Linear Unit (ReLU)*: it is defined by $f_{act}(x) = max(0, x)$. There is empirical evidence to prove an improvement in training of deep networks with respect to the hyperbolic tangent [39].

- *Leaky ReLU*: a variant of ReLU that tries to avoid the *dying ReLU* problem, where some neurons could become perpetually inactive (always produce null outputs), by setting $f_{act}(x) = 0.01x$ when $x \leq 0$. The *Parameterized ReLU* [40] is a generalization.

- *Exponential Linear Unit (ELU)*: a recent alternative to ReLU that replaces its negative part with an exponential function [41].

    The output $y_i$ of a neuron can then be defined as in Eq 2:

$$y_i = f_{act}(z_i), \tag{2}$$

where $f_{act}$ is some activation function fixed before model training.

Learning is done via *backpropagation* [38], an algorithm used to compute weight updates with respect to a loss function (usually, training error). In backpropagation, classification error on the training set is calculated with the current configuration of the network. This error is recursively propagated from the output layer to the input layer. Afterwards, the weights of the network are updated by taking into account the output of each neuron and the error calculated at each connection, so that error-inducing weights are corrected.

As a consequence of the complexity of neural networks, there are many hyperparameters (parameters fixed before learning) to tune in order to maximize the quality of the final model. The ones considered in this paper are the structure (optimized in the first step of our procedure) and the learning rate, training epochs and dropout rate (optimized in the second step). A brief definition of each one is provided below:

- Structure: it refers to the number of hidden layers and their corresponding widths (number of neurons).

- Learning rate: it represents the fraction of the measured error that is used to correct the weights of the network (thus being between 0 and 1). Smaller values make the training slower but more accurate, whereas larger values do the opposite. Finding a value that allows a faster training without missing promising error minima is fundamental.

- Number of epochs: an epoch corresponds to one forward and one backward pass of the training data through the network, in order to calculate its training error rate and adjust its weights accordingly. Too few epochs cause underfitting, while too many cause overfitting, as the number is proportional to the degree to which the class boundaries found by the network resemble the training set.

- Dropout rate: dropout [42] is a technique where random neurons are disabled with a certain probability $p$ for each training example in a batch, and then the error calculations are averaged. As a consequence, it is essentially a form of regularization [43] because, in practice, different independent sub-models with less predictive power than the whole network are combined. Again, $p$ (the dropout rate) needs to be tuned so that the final network neither overfits nor underfits.

### 3.2 Convolutional Neural Networks

The Convolutional Neural Network (CNN) is driven by the convolution operator. Although it also retains the layers found in FFNNs, fully-connected layers are no longer the main asset of the network. The convolution operator ($\star$) takes two functions and outputs a third one. The one-dimensional discrete convolution, used in this paper, is defined in Eq 3:

$$H[i] = \sum_{u=-\infty}^{\infty} F[u] \cdot G[i-u],$$ (3)

where $H$ is the result of convolving $F$ and $G$, written as $F \star G$, and $i$ points to a discrete value. Some properties of convolution are commutativity, associativity, and distributivity.

Nevertheless, a practical application to finite functions requires a finite convolution effect range. For this purpose, a number called *filter size* in the context of CNN delimits the range of the operation. Eq 4 shows the modified convolution with filter size $k$:

$$H[i] = \sum_{u=-k}^{k} F[u] \cdot G[i-u], \forall i \in \{1, \ldots, n\},$$ (4)

where $n$ is the number of elements in $G$. When applied to CNNs, the two functions passed to the convolution are interpreted as an input observation ($G$) and a filter ($F$) of a smaller size. An example can be found in Fig 4.

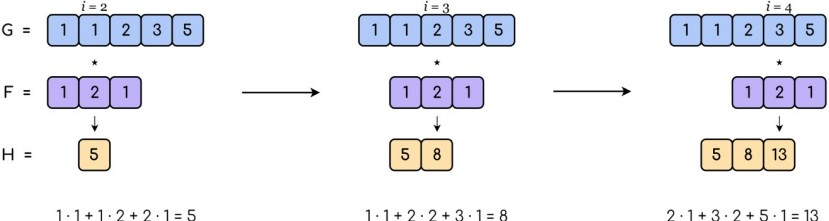

**Fig 4. 1-D convolution example.** A filter of size 3 is iteratively convolved with each element of the input (top row) to produce the output (bottom row). In this case, border values are discarded.

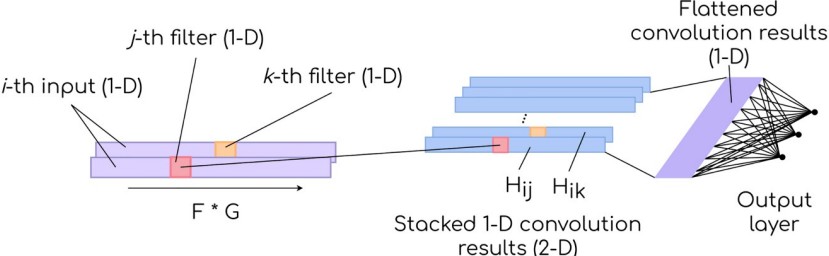

**Fig 5. 1-D CNN.** Multiple filters are applied on a one-dimensional input, creating a two-dimensional stack of convolution results. The data structure is transformed (flattened) before the fully-connected output layer.

The goal of the process is to produce a modified version of the observation by iteratively sweeping the filter along its axes. Note that, since the center of the filter must be aligned with a position of the observation, $k$ must be an odd number. This also raises the issue of whether to consider the border values or not. If they are discarded, the output is reduced in size, and otherwise either some calculations must be ignored or performed with placeholder values.

The density of FFNNs often leads to overfitting to the training data. Conversely, CNNs are sparser and more flexible. Through a series of chained convolution operations, the network builds a hierarchy of information that is progressively transformed from detailed to abstract as it goes through convolutional layers [38]. By using its own interpretation of the input data, the model extracts basic pieces of knowledge and understands them in relation to each other.

Convolution, thus, brings along new types of hidden layers:

- Convolutional layer: it is composed of multiple filters or functions that are convolved with an instance of the input data. The weights (values) of these filters are adjusted in the learning process of the model. The number of filters and the separation between consecutive convolutions along the input instance must be fixed before training.

- Pooling layer: it is a downsampling process whereby the original dimensions of the data are reduced in size, usually through a summarizing operation such as the average or the maximum of neighboring values. Since the relative location of values is more important than their absolute location, this reduction can be safely achieved. The range of application (filter size) is a hyperparameter of the summarizing function.

Fig 5 depicts the structure of a CNN with a single convolutional layer before a fully-connected output layer. Notice that the additional dimension created by stacking convolution results must be flattened, as fully-connected layers are one-dimensional.

## 3.3 Recurrent Neural Networks

The Recurrent Neural Network (RNN) can be viewed as an extension of the FFNN that works with variable-length inputs. This is made possible by a working memory in the form of a recurrent internal state. Therefore, it uses not only the information coming from each isolated value but also the surrounding context.

The most basic form of RNN involves a loop that adds a variable amount of feedback from previous values to the processing of subsequent ones. The loop can be unrolled as in Fig 6, where $x_t$ is a value of the input vector, $o_t$ is the output of the unit that has a hidden state $h_t$, and $c_f$ is the context information carried forward through the different units.

**Fig 6. Unrolled RNN loop.** Input values are at the bottom. Output values are at the top. Recurrent units lie in the middle, and are connected by a one-way context loop.

A simple recurrence scheme where $c_f$ (the context) is just $h_{t-1}$ (the hidden state from the previous unit) is described in Eqs 5 and 6, where $\theta$ represents the set of model parameters that affect the calculations. Note that there are other schemes of recurrence, such as bi-directional recurrence [44], but they are not relevant here.

$$h_t = f(h_{t-1}, x_t, \theta), \tag{5}$$

$$o_t = f(h_t, \theta), \tag{6}$$

which show that the output of the unit (that can be thought of as the activation function) is obtained solely from its hidden state. The current hidden state is calculated by using both the input and the hidden state from the previous unit. This mechanism allows the network to take into consideration already-seen inputs.

However, this architecture suffers from learning issues: as context size requirements grow, basic RNNs become unable to handle dependencies between inputs [45]. Two of the most prominent types of recurrent units that were designed to solve this problem are the Long Short-Term Memory (LSTM), first proposed in [46], and the more recent Gated Recurrent Unit (GRU), introduced in [47]. Although GRU is usually faster, as demonstrated in [48], whose notation is largely used from here on, LSTM is strictly more powerful.

LSTM units make use of five elements to tackle long-term dependencies:

- Input gate (Eq 7): it controls the extent to which new information is stored in the cell.

$$i_t = \sigma_{\log}(W^i x_t + U^i h_{t-1} + b^i). \tag{7}$$

- Forget gate (Eq 8): it decides which parts of the existing context must be forgotten based on the current input. It was added to the LSTM architecture in [49].

$$f_t = \sigma_{\log}(W^f x_t + U^f h_{t-1} + b^f). \tag{8}$$

- Output gate (Eq 9): it controls the degree to which the input to the current unit and the cell state are relevant to the output.

$$o_t = \sigma_{\log}(W^o x_t + U^o h_{t-1} + b^o). \tag{9}$$

- Cell state (Eq 10): it is where relevant contextual information is stored. As the contents do not suffer substantial changes once stored, the network is able to keep track of important details for a large number of time steps.

$$c_t = f_t \circ c_{t-1} + i_t \circ \tanh\left(W^c x_t + U^c h_{t-1} + b^c\right). \tag{10}$$

- Hidden state (Eq 11): it is where the output of the LSTM is computed.

$$h_t = o_t \circ \tanh(c_t). \tag{11}$$

In Eqs 8, 9, 10 and 11, lowercase variables are vectors (with $x_t$ being the input vector of the unit) and uppercase variables are weight matrices learned during training ($W$ matrices contain weights associated with the input and $U$ matrices contain weights for the connections coming from network units). $\sigma_{\log}$ represents logistic sigmoid functions. The operator $\circ$ is the Hadamard product (element-wise product).

The GRU is a simplified version of the LSTM. The cell element is no longer present, and only two gates are required: the reset and update gates, shown in Eqs 12 and 13. The reset gate determines how to combine the existing information with the new input, whereas the update gate decides what parts of the context should be discarded. The output of the unit is given by Eqs 14 and 15.

$$r_t = \sigma_{\log}(W^r x_t + U^r h_{t-1} + b^r). \tag{12}$$

$$z_t = \sigma_{\log}(W^z x_t + U^z h_{t-1} + b^z). \tag{13}$$

$$\tilde{h}_t = \tanh(W^h x_t + U^h(r_t \circ h_{t-1} + b^h). \tag{14}$$

$$h_t = z_t \circ h_{t-1} + (1 - z_t) \circ \tilde{h}_t. \tag{15}$$

Fig 7 depicts the operations taking place inside LSTM and GRU units according to the equations described above.

## 3.4 Genetic Algorithm for optimization

The Genetic Algorithm (GA) is an iterative search procedure that employs a population of individuals that compete on the basis of some fitness measure that drives the survival of the fittest. Through this mechanism, solutions to a problem evolve over a fixed number of generations by producing new offspring that share some similarities with their parents. More formally, the concepts illustrated in Fig 8 work together to form the structure of a GA.

The GA makes use of three sets of individuals. Solutions from the current generation are stored in the population. The set of parents from which new offspring will be created is chosen from the population. The set of offspring is combined in some way with the current population to form the population of the next generation. The population of the first generation is often filled (initialized) with random but valid solutions.

The selection operator uses the quality of the solutions to decide which are picked out from the population as parents. Candidate selection is usually randomized, so that lower-quality

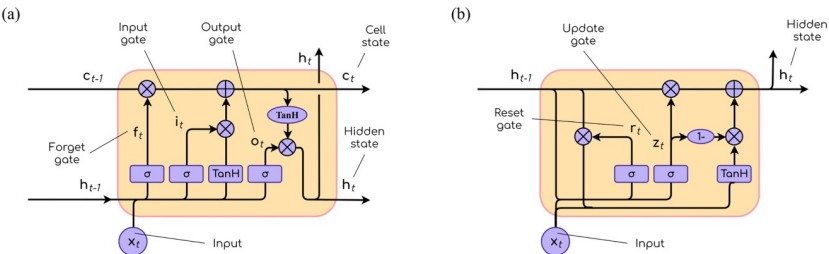

**Fig 7. LSTM and GRU schemes.** (a) Inside an LSTM unit. (b) Inside a GRU unit.

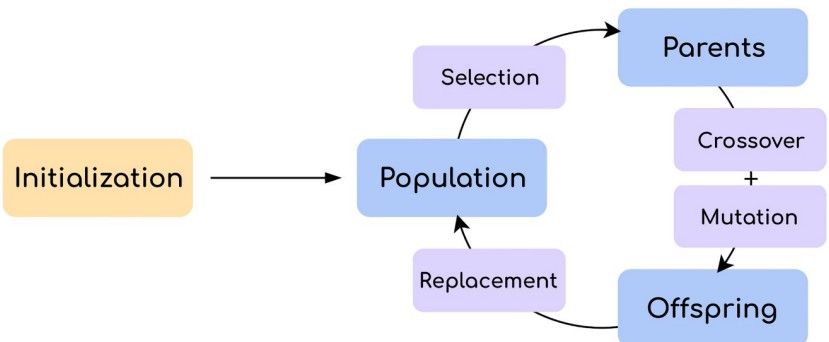

**Fig 8. Overview of a genetic algorithm.** Blue boxes represent sets of individuals. Purple boxes represent operations.

solutions have a small chance to win too. This is a mechanism to avoid excessive convergence to local optima. Selection operators are problem-agnostic, which means that they are independent from problem representation, and the role in balancing exploitation of quality solutions and exploration of alternatives has been extensively studied [50].

The crossover operator dictates the way parents, most commonly two, pass their information to new offspring. The procedure involves some degree of randomness, but it places a strong emphasis on the preservation of desirable traits. On the other hand, the mutation operator relies solely on stochastic calculations to cause unbiased changes with the purpose of introducing variability and innovation in the gene pool.

The replacement operator controls how the newly-created offspring are merged with the current population. As opposed to the real world, population size in GAs is almost always fixed, and consequently existing individuals may be replaced by newer, fitter ones. With regard to the extent of the replacement, in a steady-state paradigm only a few offspring are generated, while in a generational paradigm the whole population may be replaced by offspring. A bias towards quality (elitism) can also be introduced, for example, by merging both sets of individuals and performing a fitness-based ranking.

Individuals can encode the information of a valid solution to a problem in a variety of ways. The encoding is problem-dependent, although it is common to have multiple options whose pros and cons must be assessed. Some operators, like crossover and mutation, are in turn dependent on the chosen encoding. The fitness of an individual is evaluated by one or more fitness functions, which may check the quality of a solution in a direct or indirect way. The latter becomes essential when the computational cost of the former is prohibitive.

For more details on GAs, and on evolutionary algorithms in general, refer to [51].

Two different procedures are performed in this paper: FS and model optimization. FS is done just once, and the sets of selected features are common for all models. Model optimization consists of two sequential steps (structure optimization and learning optimization) that tackle different hyperparameter sets. In total, three types of search are performed, each by a GA specifically tailored to the task whose characteristics will be detailed in the experimental setup section. An overview of the experimental procedure can be seen in Fig 9.

### 3.5 Performance evaluation

The classification accuracy of the models is evaluated according to two different performance measures: cross-validation accuracy (to guide the optimization algorithms) and the Kappa coefficient (to compare final models to each other):

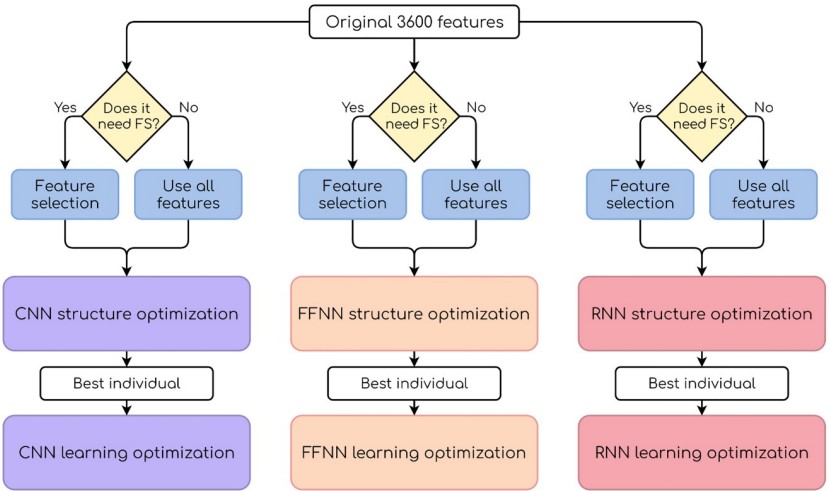

**Fig 9. General scheme of the experiments.** The parameters from the best individual in the first optimization step are used in the second step.

- Cross-validation accuracy: the training set is split into $n$ sections, iteratively using $n-1$ of them for model training and leaving the remaining one out for testing. The final value is the arithmetic mean of the $n$ test accuracies. The range of possible values is [0, 1].

- Cohen's Kappa accuracy [52]: similar to the test set accuracy, but also taking into account the possibility of classifying correctly by chance. It is computed as:

$$\kappa = \frac{p_0 - p_c}{1 - p_c},\qquad(16)$$

where $p_0$ is equivalent to the test set accuracy, and $p_c$ is the sum of the probabilities of random agreement for all possible classes. Its values lie within the range [−1, 1].

### 3.6 Statistical analysis

The assessment of differences in performance between alternatives can benefit from statistical analysis, as it allows for safer extrapolation. In the present paper, we make use of two statistical approaches: Null Hypothesis Significance Testing (NHST) and Bayesian Testing.

Three alternatives (one for each type of neural network) will be involved in the final comparison. The general issue of whether there are any differences between the alternatives has to be addressed before carrying out more specific, pairwise comparisons. For this purpose, the non-parametric Friedman test [53, 54] will be employed.

If the Friedman test rejects the null hypothesis of equivalence ($p < 0.05$), post-hoc tests will be performed to find pairwise differences. However, since it can be inconsistent in some cases [55], we will use instead the Wilcoxon Signed-Rank test, as in [56]. Moreover, the Family-Wise Error Rate (FWER) must be taken into account [57]. The FWER is the probability of incorrectly rejecting the null hypothesis (type I error, or false positive) when carrying out multiple comparisons. A number of $p$-value corrections have been proposed to solve this problem, among them the Holm method [58] used here. When applicable, adjusted $p$-values will be reported in the experimental results section.

Nonetheless, NHST has several known shortcomings. For instance, it does not provide probabilities for the tested hypotheses. Furthermore, as perfect equivalence is highly unlikely, strict null hypotheses are always false in practice, which can detract from the interpretation of the results. In addition, no conclusions can be drawn when NHST fails to reject the null hypothesis, and even the threshold for the *p*-value is as customary as it is arbitrary. Lastly, NHST does not directly answer the question of whether the experimental results show differences between alternatives, but rather the question of how likely the obtained results are assuming that the null hypothesis is true. For these reasons (among others, all described in [59]), we complement NHST with Bayesian Testing, which is able to overcome the aforementioned limitations.

The Bayesian Signed-Rank test is suitable for pairwise comparisons over multiple datasets and will help to ascertain pairwise differences between alternatives. This test obtains the distribution of a certain parameter $z$ under the assumption that it is a Dirichlet distribution. Let $A$ and $B$ be the sets of quality measurements of two algorithms. The distribution is calculated by counting the occurrences in the available data of $b - a > 0$, $b - a < 0$, and $b - a \approx 0$, where $a$ and $b$ are elements of $A$ and $B$. The last one represents the Region of Practical Equivalence (rope), which contains non-significant differences in the range $[\text{rope}_{min}, \text{rope}_{max}]$. The Dirichlet distribution is built from these results, and is then sampled to obtain triplets of the form shown in Eq 17:

$$[P(z < \text{rope}_{min}) = P(b - a < 0), P(z \in \text{rope}) = P(b - a \approx 0), P(z > \text{rope}_{max}) = P(b - a > 0)]. \quad (17)$$

Every triplet, if interpreted as barycentric coordinates, can be represented as a point contained within the boundaries of an equilateral triangle. Each value of a triplet is associated with a different vertex of the triangle. The location of the point described by a triplet is determined as follows: for each of its three values in isolation, the distance between its associated vertex and the point (in the direction of the opposing side) is proportional to the value. Thus, the higher the value, the closer the point will be to that vertex. A heatmap can be created by doing this for every sampled triplet. Since there are three possibilities for $b - a$ (negative, positive, and equivalent), each one is associated with one vertex (left, right, and top, respectively). Broadly speaking, the heatmap can be interpreted as: the closer the point cloud is to a vertex, the higher the probability of that option being true. The heatmap is the visual representation of the probabilities calculated by the Bayesian test for the dominance of either algorithm and their equivalence. Fig 10, which exemplifies this, shows a point cloud mostly within the upper region and partly crossing into the lower right region, which means that a tie is expected in a majority of cases, but otherwise the advantage is often for algorithm *B*.

## 4 Experimental results and discussion

The contents of this section are organized as follows: firstly, the experimental setup is described, including software, hardware, and parameters for the GAs; secondly, the effects of FS and the subsequent hyperparameter optimization procedure are discussed in a separate section for each type of neural network (CNN, FFNN, and RNN); thirdly, the three optimized alternatives are compared in terms of classification accuracy and energy-time cost.

### 4.1 Experimental setup

The code was written in *Python 3.4.9*. A number of state-of-the-art machine learning libraries are available for this language, some of which are used here: *Scikit-Learn 0.19.2* [60], *NumPy 1.14.5* [61], and *TensorFlow 1.10.1* [62] as a backend for *Keras 2.2.2* [63]. The Friedman test

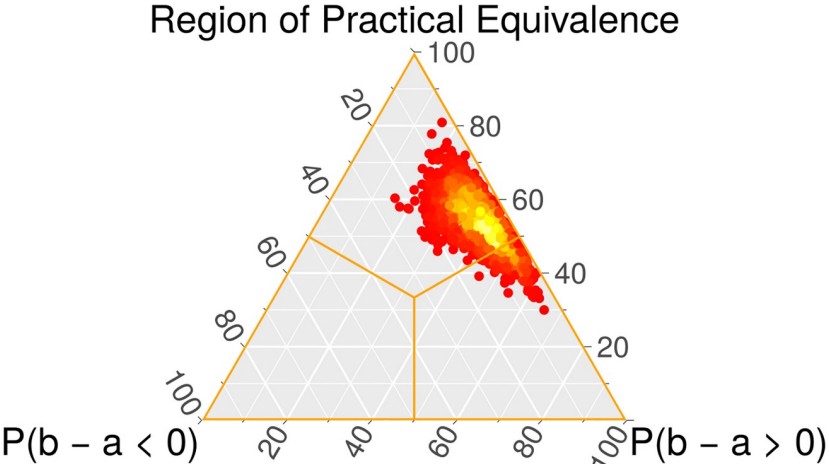

**Fig 10. Example of heatmap from the Bayesian Signed-Rank test.** The probabilities associated with the heatmap are: $P(b - a < 0) = 0.078$, $P(b - a > 0) = 0.382$, and $P(b - a \approx 0) = 0.540$.

and the Wilcoxon test are provided by the R library `scmamp` [64]. Bayesian tests are implemented in the R library `rNPBST` [65].

The experiments were carried out on a four-node cluster, of which two nodes equipped with an NVIDIA TITAN Xp GPU were used. The energy-time measurements were recorded on Node 4, which has the following hardware specifications:

- Two Intel® Xeon® E5-2620 v4 @ 2.10 GHz: 16 cores/32 threads and a Thermal Design Power (TDP) of 85W.

- NVIDIA TITAN Xp: 3, 840 CUDA cores, 12 GB of GDDR5X RAM memory, and a TDP of 250W.

As previously stated, GAs are used to perform three different searches: one in FS, and two in the two-step hyperparameter optimization procedure (structure and learning). They have some details in common: selection is done by binary tournament, replacement is elitist, and fitness is evaluated by 5-fold cross-validation. Note that the model used in cross-validation to evaluate the folds depends on the application. For FS, Logistic Regression is a very light model that allows for many fitness evaluations, but the choice is free. On the contrary, for hyperparameter optimization the only option is the particular type of neural network that is being optimized. The genetic parameters that are specific to each application are presented below.

The encoding of the individuals in FS is straightforward: one binary value for each of the 3, 600 features, where 1 means the feature is active and 0 means inactive. This enables the use of simple yet powerful operators: Uniform crossover, which ensures all common elements in the parents are passed on to the offspring and randomly assigns the rest, and Bit-flip mutation, which inverts a randomly chosen binary value. The first population is initialized by sampling amounts of active features from a uniform distribution between 1 and 3, 600. See Table 1 for a summary. Before performing any hyperparameter optimization procedure, the use of FS will be discussed for each type of network by taking into account its properties.

The FS procedure is performed only once, and the selected features for each dataset are common to any neural network that finally makes use of them. For dataset 104, 25 features from a total of 3, 600 were selected. For datasets 107 and 110, 28 features were selected.

**Table 1. Summary of parameter values for the GA used in the FS procedure.**

| Parameter | Value |
|---|---|
| Encoding | Binary (one bit for each feature) |
| Population | 1000 individuals |
| Generations | 100 generations |
| Crossover | Uniform |
| Mutation | Bit-flip |
| Constraints | Maximum of 30 features out of 3, 600 |

The configuration of the GA for each step of the hyperparameter optimization procedure is different for each type of network. However, to ensure a minimum degree of fairness, the amount of individuals (40) and generations (10) is common. The method used to initialize the first population is also common: the values are sampled from a normal distribution centered at the middle of the allowed ranges.

The hyperparameters considered for CNN layers are filter size and number of filters (width), while for FFNN and RNN layers only their width is tuned. The structures of CNN and RNN will have fixed depth due to complexity and time constraints, and will have one hidden layer (convolutional and GRU, respectively) whose parameters will be optimized. Moreover, for the case of CNN, the ordering of a variable amount of heterogeneous layers, such as convolutional and pooling, through crossover and mutation can prove cumbersome. FFNN, on the other hand, is allowed to have up to two hidden layers (of uncapped width to preserve layer size ratios found by the GA). Although the resulting networks will be shallow, the existing literature suggests that deeper networks are harder to calibrate with the often small EEG datasets [13]. Finally, each alternative uses the same activation function throughout the experiments: *ReLU* for CNN and RNN, and *ELU* for FFNN (we observed that FFNN got stuck in learning with *ReLU*).

Given that CNN and RNN structures are encoded by fixed-size individuals, the genetic operators can be similar to those applicable to real-coded optimization problems. The Single-point crossover creates a cutoff point in both parents so that the offspring inherits the first part from one parent and the second part from the other. The Gaussian mutation multiplies the individual by a value sampled from a normal distribution centered at 1. The depth variability of FFNN structures is reflected in the operators. In the Midpoint crossover, the offspring inherits one half of the structure from each parent. In the Single-layer and scaling mutations, a Gaussian modification is applied to one layer or the whole network, respectively. Table 2 contains a summary of the parameter values.

**Table 2. Summary of parameter values for the GA used in CNN, FFNN, and RNN structure optimization.**

| Parameter | Value | | |
|---|---|---|---|
| | CNN | FFNN | RNN |
| Encoding | Real (fixed size) | Real (variable size) | Real (fixed size) |
| Population | 40 individuals | | |
| Generations | 10 generations | | |
| Crossover | Single-point | Midpoint | Single-point |
| Mutation | Gaussian | Single-layer/scaling | Gaussian |
| Constraints | 250 filters, size 19 | 2 hidden layers | 60 neurons per layer |

**Table 3. Summary of parameter values for the GA used in CNN, FFNN, and RNN learning optimization.**

| Parameter | Value | | |
|---|---|---|---|
| | **CNN** | **FFNN** | **RNN** |
| **Encoding** | Real (fixed size) | | |
| **Population** | 40 individuals | | |
| **Generations** | 10 generations | | |
| **Crossover** | Single-point | | |
| **Mutation** | Gaussian | Gaussian/dropout | Gaussian |
| **Constraints** | 200 training epochs | | |

The second optimization step is much more homogeneous than the first. The learning rate and the number of epochs are tuned for all three types of networks, and the dropout rate is also tuned for FFNN. The individuals are real-coded and fixed in size. Because of their similarities with the previous step for CNN and RNN, the Single-point and Gaussian operators are used again, with the only novelty being a Gaussian mutation specifically for the dropout rate (when mutation is needed, one of the two options is chosen at random). The summary of this step can be found in Table 3.

The structure constraints of CNN and RNN, which are the two most expensive alternatives, have been set with a balance between freedom for the GA and feasibility in terms of computation times in mind. The limit on the amount of training epochs is the same across the 3 alternatives to make the comparison fairer, as it is a less architecture-specific hyperparameter. As a final note, we highlight that the experimental results have been averaged over 15 executions.

## 4.2 CNN optimization

The design of the CNN allows it to perform implicit FS through the application of convolution filters that find meaningful local patterns in the data. Intuitively, prior FS done by a different algorithm could hinder the process by removing features that could be important in relation to others. This could cancel any benefits brought by dimensionality reduction, such as a decrease in overfitting. In particular, our FS procedure evaluates candidate feature subsets with Logistic Regression, whose optimally informative features do not necessarily coincide with those of CNN. To illustrate this, Table 4 compares average test-set Kappa values of 15 CNN models trained using selected features (CNN + FS) and 15 CNN models using all features (CNN + noFS). As can be observed, CNN combined with FS appears to perform significantly worse.

In this case, the choice of not using FS seems reasonable (and is also supported by our previous paper [32]), since there could be a dramatic drop in quality. Hyperparameter optimization is then performed on CNN with 3, 600 input features (CNN + noFS). Table 5 compares the unoptimized results with the optimized results (CNN + noFS + OPT), again averaged over 15 executions. Fig 11 offers a visual comparison.

**Table 4. Test-set Kappa for CNN with FS (CNN + FS) and without FS (CNN + noFS).** SD: Standard Deviation. Trained with 60 epochs, learning rate of 0.1, one hidden convolutional layer of 130 filters of size 5, and ReLU activation.

| Subject | CNN + noFS | | CNN + FS | |
|---|---|---|---|---|
| | Avg. ± SD | Best | Avg. ± SD | Best |
| **104** | 0.7096 ± 0.0130 | 0.7295 | 0.6210 ± 0.0098 | 0.6353 |
| **107** | 0.7099 ± 0.0125 | 0.7306 | 0.5414 ± 0.0114 | 0.5632 |
| **110** | 0.6185 ± 0.0170 | 0.6376 | 0.5606 ± 0.0180 | 0.5871 |

**Table 5. Test-set Kappa for CNN with optimization (CNN + noFS + OPT) and without optimization (CNN + noFS).** SD: Standard Deviation.

| Subject | CNN + noFS | | CNN + noFS + OPT | |
|---|---|---|---|---|
| | Avg. ± SD | Best | Avg. ± SD | Best |
| 104 | 0.7096 ± 0.0130 | 0.7295 | 0.7396 ± 0.0055 | 0.7546 |
| 107 | 0.7099 ± 0.0125 | 0.7306 | 0.7218 ± 0.0121 | 0.7392 |
| 110 | 0.6185 ± 0.0170 | 0.6376 | 0.6527 ± 0.0088 | 0.6713 |

The original unoptimized CNN models without FS already offered decent performance for datasets 104 and 107, but the optimized models offer an improvement of roughly 0.03 and 0.01 average points. The quality for dataset 110, while more modest, shows the biggest improvement: roughly 0.035 average points. Peak Kappa values also exhibit marked increases in all datasets.

## 4.3 FFNN optimization

Admittedly, networks with densely-connected sequences of layers as FFNN have considerable function approximation power. Nevertheless, the amount of learnable parameters (one per connection between two given neurons) can be inordinate for small datasets, as the network becomes unable to ignore trivial details and suffers from overfitting. With 3, 600 features and only 178 training examples in our datasets, this situation is foreseeable. Aside from establishing preemptive size limits in the tuning procedure, FS can also be helpful in removing uninteresting features. As can be seen in Table 6, where we compare 15 FFNNs using FS to 15 FFNNs without using FS, reduced feature sets seem to alleviate overfitting on average. Moreover, quality appears to stabilize, according to the observed differences in standard deviation.

For the reasons discussed above, considering only selected features is probably the best option in this case. In consequence, Table 7 and Fig 12 compare 15 FFNN + FS models to their 15 optimized counterparts (FFNN + FS + OPT). The results show notably superior test-set Kappa values in favor of FFNN + FS + OPT: about 0.03, 0.04, and 0.035 average points for datasets 104, 107, and 110, respectively. Peak values also show significant gains.

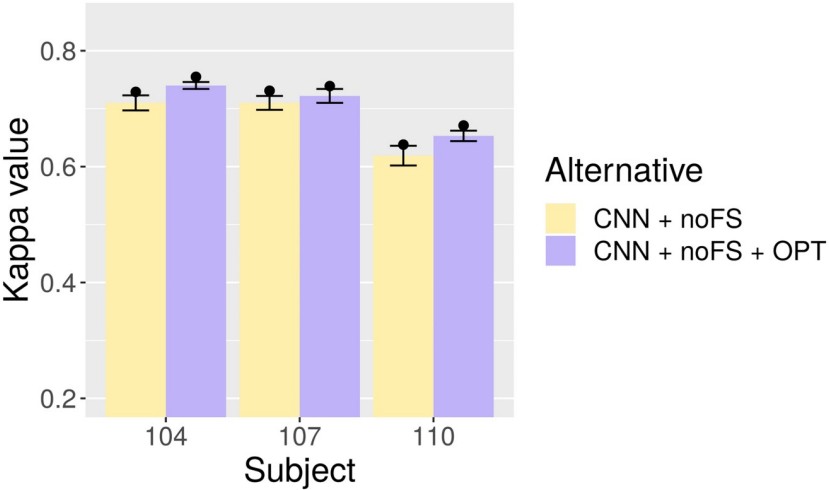

**Fig 11. Test-set Kappa for CNN with and without optimization.** Bars represent averages, points mean peak performance, and lines delimit the range of the standard deviation.

**Table 6. Test-set Kappa for FFNN with FS (FFNN + FS) and without FS (FFNN + noFS).** SD: Standard Deviation. Trained with 60 epochs, learning rate of 0.1, one hidden fully-connected layer of size 100, and ELU activation.

| Subject | FFNN + noFS | | FFNN + FS | |
|---|---|---|---|---|
| | Avg. ± SD | Best | Avg. ± SD | Best |
| 104 | 0.4800 ± 0.1376 | 0.5870 | 0.6612 ± 0.0121 | 0.6784 |
| 107 | 0.3742 ± 0.1769 | 0.5777 | 0.5242 ± 0.0184 | 0.5467 |
| 110 | 0.3979 ± 0.2024 | 0.5783 | 0.5950 ± 0.0266 | 0.6379 |

**Table 7. Test-set Kappa for FFNN with optimization (FFNN + FS + OPT) and without optimization (FFNN + FS).** SD: Standard Deviation.

| Subject | FFNN + FS | | FFNN + FS + OPT | |
|---|---|---|---|---|
| | Avg. ± SD | Best | Avg. ± SD | Best |
| 104 | 0.6612 ± 0.0121 | 0.6784 | 0.6901 ± 0.0147 | 0.7289 |
| 107 | 0.5242 ± 0.0184 | 0.5467 | 0.5630 ± 0.0107 | 0.5796 |
| 110 | 0.5950 ± 0.0266 | 0.6379 | 0.6299 ± 0.0109 | 0.6464 |

## 4.4 RNN optimization

The effects of combining RNN with FS are not as clear as for CNN and FFNN. On one hand, the ability of RNN to store context information could be impaired by external FS. On the other hand, the use of 3, 600 features could greatly increase training time and risk of overfitting, because each GRU unit performs several operations and the architecture of the network is still fully-connected. Table 8, where the two alternatives are compared on two sets of 15 RNNs, suggests that neither of them has a decisive advantage: each one is noticeably better in one dataset, and both are approximately equivalent in the third. In light of this, it may be more interesting to find out the potential quality of the faster alternative (FS). Therefore, hyperparameter optimization will make use of the selected feature sets.

Table 9 and Fig 13 contain the comparison between unoptimized RNN (RNN + FS) and optimized RNN (RNN + FS + OPT). The latter achieves improvements in average Kappa

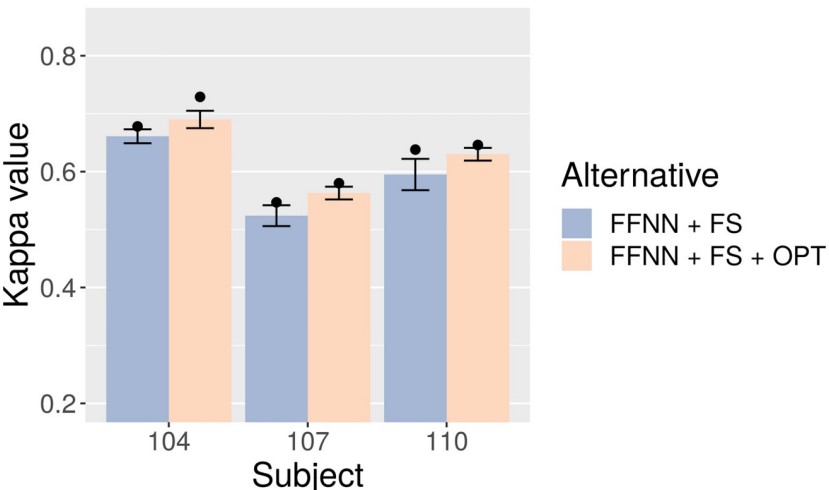

**Fig 12. Test-set Kappa for FFNN with and without optimization.** Bars represent averages, points mean peak performance, and lines delimit the range of the standard deviation.

**Table 8. Test-set Kappa for RNN with FS (RNN + FS) and without FS (RNN + noFS).** SD: Standard Deviation. Trained with 60 epochs, learning rate of 0.1, one hidden GRU layer of size 16, and ReLU activation.

| Subject | RNN + noFS | | RNN + FS | |
|---|---|---|---|---|
| | Avg. ± SD | Best | Avg. ± SD | Best |
| 104 | 0.5829 ± 0.0153 | 0.6103 | 0.6326 ± 0.0225 | 0.6779 |
| 107 | 0.6316 ± 0.0127 | 0.6470 | 0.5249 ± 0.0129 | 0.5546 |
| 110 | 0.5005 ± 0.0319 | 0.5533 | 0.4965 ± 0.0225 | 0.5367 |

**Table 9. Test-set Kappa for RNN with optimization (RNN + FS + OPT) and without optimization (RNN + FS).** SD: Standard Deviation.

| Subject | RNN + FS | | RNN + FS + OPT | |
|---|---|---|---|---|
| | Avg. ± SD | Best | Avg. ± SD | Best |
| 104 | 0.6326 ± 0.0225 | 0.6779 | 0.6697 ± 0.0216 | 0.7125 |
| 107 | 0.5249 ± 0.0129 | 0.5546 | 0.5704 ± 0.0089 | 0.5799 |
| 110 | 0.4965 ± 0.0225 | 0.5367 | 0.5901 ± 0.0206 | 0.6208 |

values of about 0.035, 0.045 in datasets 104 and 107, but most notably of nearly 0.1 in dataset 110 (where it severely lagged behind the other two alternatives).

## 4.5 Accuracy comparison

In the previous three sections, the results of the hyperparameter optimization procedure have been analyzed within the context of each type of network. In this regard, the proposed procedure has proven to be able to enhance the accuracy of the models. However, a comparison of the three alternatives is mandatory, both in terms of accuracy and energy-time cost.

Table 10 and Fig 14 hold the test-set Kappa values obtained by each optimized alternative in their corresponding sections. CNN is superior to FFNN and RNN in all datasets, and

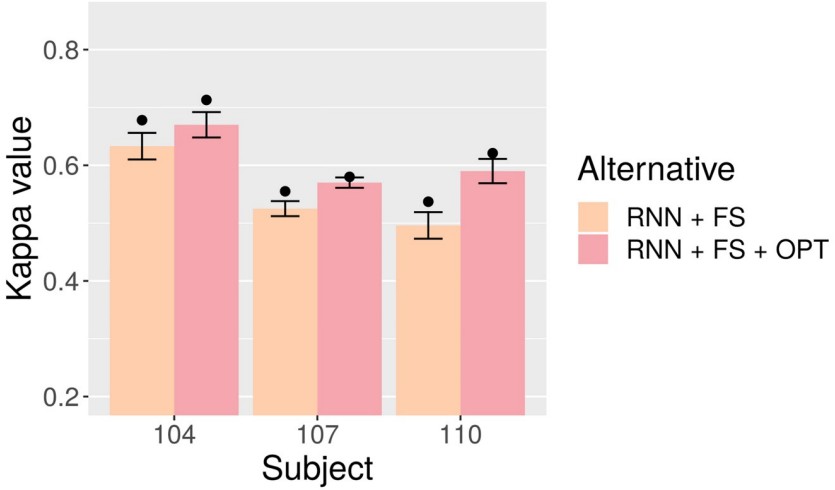

**Fig 13. Test-set Kappa for RNN with and without optimization.** Bars represent averages, points mean peak performance, and lines delimit the range of the standard deviation.

**Table 10. Test-set Kappa comparison for CNN + noFS + OPT, FFNN + FS + OPT, and RNN + FS + OPT.** SD: Standard Deviation.

| Subject | Measure | CNN + noFS + OPT | FFNN + FS + OPT | RNN + FS + OPT |
|---|---|---|---|---|
| 104 | Avg. ± SD | 0.7396 ± 0.0055 | 0.6901 ± 0.0147 | 0.6697 ± 0.0216 |
| | Best | 0.7546 | 0.7289 | 0.7125 |
| 107 | Avg. ± SD | 0.7218 ± 0.0121 | 0.5630 ± 0.0107 | 0.5704 ± 0.0089 |
| | Best | 0.7392 | 0.5796 | 0.5799 |
| 110 | Avg. ± SD | 0.6527 ± 0.0088 | 0.6299 ± 0.0109 | 0.5901 ± 0.0206 |
| | Best | 0.6713 | 0.6464 | 0.6208 |

especially in 107, which is remarkable since in previous papers using the same data it is usual for proposals to achieve similar performance in datasets 104 and 107.

In contrast, FFNN and RNN have much smaller differences, with FFNN scoring higher in two of the three datasets. When merely looking at the numbers, FFNN is slightly better than RNN, but we will later make use of statistical testing to determine the extent of all the observed differences.

Concerning the underlying reasons for the experimental results, the variability of the results has two primary sources: dataset potential and model performance. In relation to the former, it is important to remember that the instances of the datasets correspond to human attempts at imagining limb movements. Some subjects are more skilled at this task than others, which affects the amount of useful information present in the EEG patterns. In addition, EEG technology suffers from artifacts in the recording process that can only be partially mitigated and add uncertainty to the data.

The observed differences in model performance can be largely attributed to the characteristics of each network. In particular, CNN appears to be the best here at finding key relationships between features. The convolution operator allows CNN to focus on relative changes rather than absolute ones, and this flexibility is a valuable asset when dealing with inconsistent inputs. On the other hand, FFNN has a tendency to overfit the training examples due to its more rigid architecture, but in view of the experimental results, FS probably alleviates this problem by

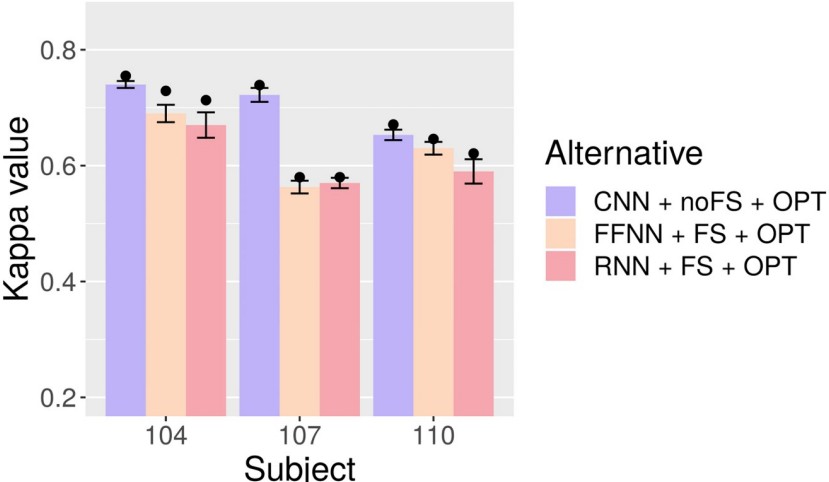

**Fig 14. Test-set Kappa for RNN with and without optimization.** Bars represent averages, points mean peak performance, and lines delimit the range of the standard deviation.

**Table 11. Holm-adjusted *p*-values for the pairwise Wilcoxon post-hoc tests.** Non-significant differences ($p > 0.05$) in bold.

| Alternative | FFNN + FS + OPT | RNN + FS + OPT |
|---|---|---|
| CNN + noFS + OPT | $p = 7.78 \cdot 10^{-9}$ | $p = 7.78 \cdot 10^{-9}$ |
| FFNN + FS + OPT | | $p = 4.34 \cdot 10^{-4}$ |

**Table 12. Probabilities (rounded) given by the Bayesian Signed-Rank test for the pairwise comparisons.** For each pair, the first method corresponds to *a*, and the second method to *b*.

| Comparison | P($b - a < 0$) | P($b - a \approx 0$) | P($b - a > 0$) |
|---|---|---|---|
| RNN + FS + OPT vs. FFNN + FS + OPT | 0.079 | 0.334 | 0.587 |
| FFNN + FS + OPT vs. CNN + noFS + OPT | 0.000 | 0.038 | 0.962 |
| RNN + FS + OPT vs. CNN + noFS + OPT | 0.000 | 0.022 | 0.978 |

removing unnecessary features that misguide the network. RNN is able to extract knowledge by leveraging context, although it might be more effective if learning from the 3, 600 features were computationally viable.

Regarding the statistical analysis, the Friedman test to detect significant differences rejects the null hypothesis ($p = 1.99 \cdot 10^{-15}$, $\chi^2 = 67.744$), which means that further pairwise tests are appropriate. This is not surprising, though, seeing how CNN stands out from the other two. Table 11 reports the Wilcoxon post-hoc *p*-values.

For a significance level of 5% all pairwise null hypotheses are rejected, which means the test finds the three methods sufficiently different from one another. This result is especially relevant for the comparison of FFNN and RNN, whose average accuracies are relatively close.

The Bayesian Signed-Rank test supports these conclusions from another point of view. Table 12 reports the pairwise probabilities of dominance and tie (depicted in the heatmaps of Fig 15). The test confirms the advantage of CNN over FFNN and RNN. A probability of exactly 0.0 could seem extreme, but it is reasonable: unlike NHST, where the probability of obtaining the existing data is calculated assuming that the null hypothesis is true, Bayesian tests compute probabilities based on the data; if the data contains no observations where CNN is worse than FFNN or RNN, the output will change accordingly.

As for FFNN against RNN, the test assigns a fairly high probability to FFNN being better (see the location of the point cloud), which is consistent with the previous NHST results. Although equivalence is not fully discarded either, the sum of these two probabilities renders an advantage in favor of RNN highly unlikely.

As a consequence of the probabilities, the heatmap for FFNN and RNN contains a point cloud that is close to the FFNN vertex but is also reasonably near the Region of Practical Equivalence, and the heatmaps involving CNN have all points at maximum distance from the left vertex.

Lastly, given the existence of previous work on these datasets, it is possible to make a comparison with other approaches. Table 13 compares Deep Belief Networks (DBNs) from [66] to CNN + noFS + OPT. The results of the two are comparable, with small differences in favor of either one depending on the dataset. What is interesting about this comparison, though, is that the DBNs needed six fully-connected layers to produce these results, while the CNNs only needed three layers (input, convolutional, and output).

### 4.6 Energy-time comparison

When dealing with computationally intensive algorithms, the quality of the solutions they provide is not the only matter of interest. Time constraints and hardware limitations are

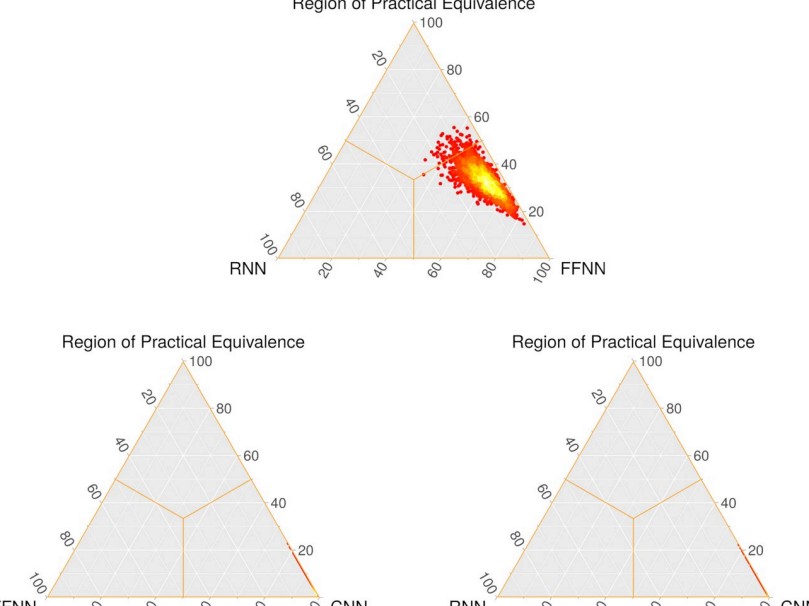

**Fig 15. Heatmaps for the Bayesian pairwise comparisons of the three networks.** a) RNN + OPT and FFNN + OPT. b) FFNN + OPT and CNN + OPT. c) RNN + OPT and CNN + OPT The left vertex corresponds to $b-a < 0$, the right vertex to $b-a > 0$, and the top vertex to $b-a \approx 0$. Sample size of 2, 000 points. Red means lower density, while brighter colors mean higher densities.

commonplace, and thus computational costs must also be factored in when assessing available options. Moreover, energy-saving has become a relevant issue in computer science and engineering. Besides the economic and environmental reasons, energy is an important concern to reach exascale performance, which would be required in many big data applications that need high performance computing for processing the corresponding neural network models (a 2010 report by the US Department of Energy estimates the annual power cost of operating an exascale system implemented with current technology to be about 2.5 billion dollars per year) [67]. With this in mind, energy-time measurements have been obtained from Node 4 (see experimental setup). The averages are displayed in Table 14.

At first glance, each model is readily distinguishable from the others. Computing time is roughly doubled from FFNN to RNN and again from RNN to CNN. Energy consumption follows a similar trend from FFNN to RNN, but the pattern is broken from RNN to CNN: the latter consumes three times more energy than the former. This can be observed in Fig 16, where the energy and time bars of CNN have been represented with the same height to facilitate the comparison.

**Table 13. Test-set Kappa values for CNN + noFS + OPT and DBN-opt.** SD: Standard Deviation.

|  | CNN + noFS + OPT | | DBN-opt | |
| --- | --- | --- | --- | --- |
| Subject | Avg. ± SD | Best | Avg. ± SD | Best |
| 104 | 0.740 ± 0.006 | 0.755 | 0.733 ± 0.011 | 0.750 |
| 107 | 0.722 ± 0.012 | 0.739 | 0.723 ± 0.007 | 0.733 |
| 110 | 0.653 ± 0.009 | 0.671 | 0.672 ± 0.008 | 0.683 |

**Table 14. Average energy-time behavior for the three optimized networks.** SD: Standard Deviation.

| Alternative | Time (s) ± SD | Energy consumed (W · h) ± SD |
|---|---|---|
| CNN + noFS + OPT | 20, 141 ± 4, 005 | 1, 608 ± 390 |
| FFNN + FS + OPT | 6, 552 ± 122 | 301 ± 5 |
| RNN + FS + OPT | 11, 492 ± 484 | 544 ± 20 |

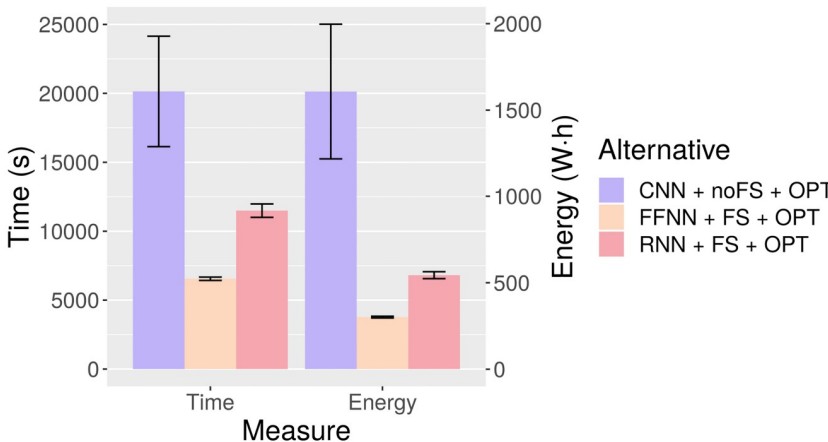

**Fig 16. Energy-time behavior for the three optimized networks.** Bars represent averages and lines delimit the range of the standard deviation.

This phenomenon can be explained by looking at Fig 17. The energy consumed by a node is determined by its running time and instantaneous power (the energy consumption is equal to the area under the instantaneous power curve along the running time). FFNN and RNN consume less instantaneous power than CNN, and therefore their total energy consumption is

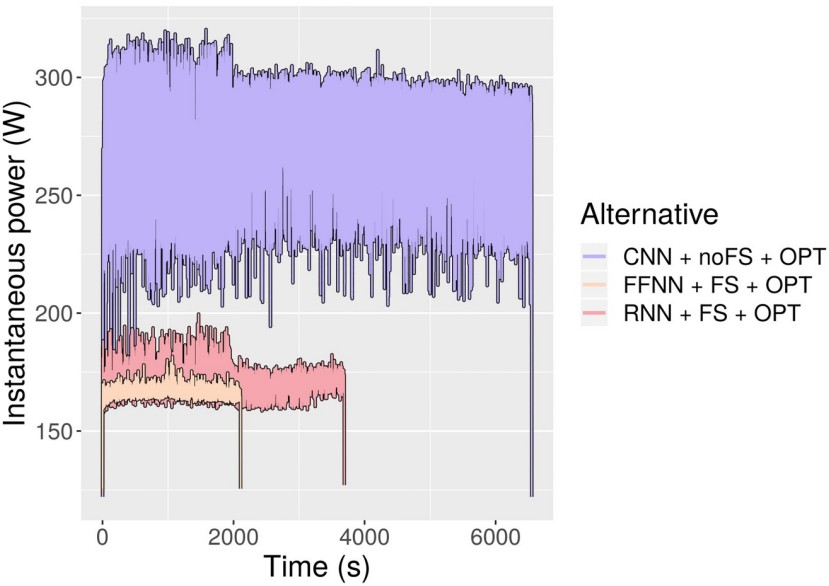

**Fig 17. Instantaneous power for the three optimized networks.**

certain to be even lower in proportion than their running time. In fact, even at equal running times they would consume less energy than CNN. Conversely, the gap in instantaneous power between FFNN and RNN is not significant enough to produce vastly different energy-time ratios. Regarding the underlying cause of the differences in instantaneous power, it is probably related to the degree to which each type of network takes advantage of GPU cores. This also includes the fact that only CNN uses 3, 600 features, which require many more parallel calculations than the reduced subsets.

### 4.7 Relationship between model quality and complexity

As shown in the previous section, the varying levels of complexity of different architectures such as convolutional, recurrent, or feed-forward have a great impact on training times. Nonetheless, variation can also exist within the same type of network. For instance, the amount of epochs needed to converge to a local optimum linearly affects training costs, and the size of the network (e.g. amount of filters in CNN, neurons in FFNN, and recurrent units in RNN) is also a major contributor to computational loads.

Therefore, it is worth looking at the characteristics of the solutions found by the optimization procedure. In Fig 18 the test-set Kappa values are shown against the total amount of convolutional filters of the 15 CNN for each dataset. It is possible to observe that there is not necessarily a well-defined correlation between quality and size of the network. While the best overall results are found in the 104 cluster, which is further than the other two in the complexity axis, the best solution for dataset 104 is closer to the middle within that context. A closer look at the 107 and 110 clusters also supports this, as they are rather uniformly distributed but with more peaks around the middle. Training epochs do not appear to follow an identifiable pattern. Although more data points would increase the confidence in the inference, it could be argued that favoring simpler architectures in the optimization procedure could still allow for similar quality at a reduced cost.

In Fig 19, the total amount of neurons in the 15 FFNN is depicted against their corresponding test-set Kappa values. As opposed to Fig 18, this time the network configurations for dataset 104 have a marked tendency towards simplicity. For the other two datasets there is more dispersion, but the overall conclusion seems to be that substantially cheaper structures with competitive quality are feasible for FFNN as well.

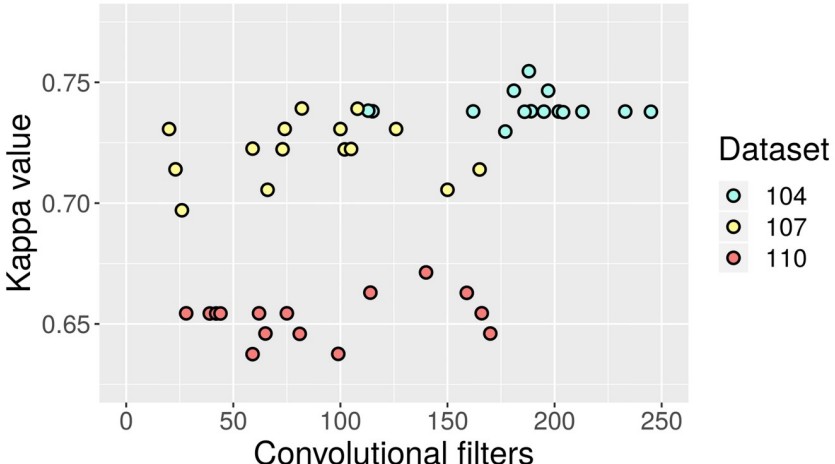

**Fig 18. Test-set Kappa values against amount of convolutional filters in CNN.**

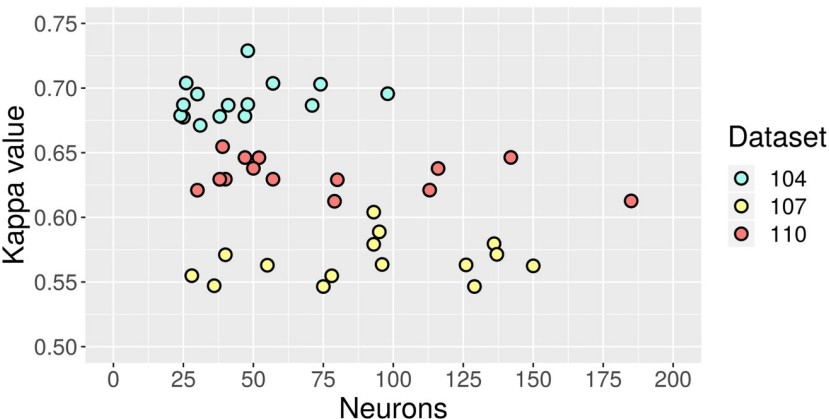

**Fig 19. Test-set Kappa values against amount of neurons in FFNN.**

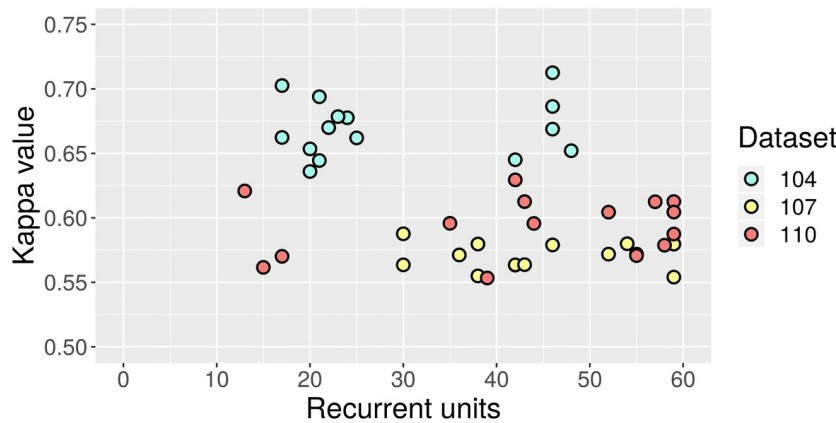

**Fig 20. Test-set Kappa values against amount of recurrent units in RNN.**

Regarding RNN, the data points in Fig 20 are arranged in a more unique way. The results for dataset 104 are split into two groups with clearly distinct amounts of recurrent units. In this case, a more complex RNN achieves the best test-set Kappa value, though it does not have a considerable advantage over the second-best. The 107 and 110 clusters overlap, but there is no apparent correlation between quality and complexity either.

The training epochs for each of the models are not shown in the previous figures to prevent clutter. However, upon examination, no straightforward link with network size was found, which is probably caused by the interdependence between learning rates and epochs. What should be highlighted, nevertheless, is that the optimization process found RNN models that learned in substantially fewer epochs (see Table 15).

**Table 15. Average training epochs needed by each alternative.** SD: Standard Deviation.

| | Average training epochs ± SD (by test subject) | | |
|---|---|---|---|
| **Alternative** | **104** | **107** | **110** |
| **CNN** | 76.46 ± 34.98 | 114.80 ± 41.76 | 84.87 ± 48.17 |
| **FFNN** | 75.07 ± 18.13 | 82.27 ± 12.52 | 78.53 ± 12.12 |
| **RNN** | 11.46 ± 4.25 | 14.27 ± 7.99 | 15.13 ± 9.72 |

For more details, the complete list of final models with their tuned hyperparameters can be found in S1 Table (CNN), S2 Table (FFNN), and S3 Table (RNN).

## 5 Conclusions

Classification in high-dimensional spaces poses by itself a challenge due to the curse of dimensionality. The difficulty is further amplified when training data is scarce in proportion, as is usual in BCI applications. EEG classification, the focus of this paper, is no exception. In addition to feature reduction procedures, such as FS, model hyperparameters must be carefully chosen in order to avoid overfitting. This is especially true of neural networks, whose excellent function approximation capabilities can be a double-edged sword.

In this paper, we propose a new procedure that divides hyperparameter search into two steps, each of them involving a certain group of related hyperparameters, in an attempt to reduce the search space and decrease computation time. The procedure is performed on three types of neural networks (namely: CNN, FFNN, and RNN) that are then evaluated and compared in terms of classification accuracy and energy-time consumption. The suitability of FS for each particular case is also discussed. In this regard, we have found that CNN achieves the best accuracy overall and probably works better without FS. FFNN and RNN, which do use FS, do not reach the same standards, but not using FS on them does not appear to guarantee improvements. However, RNN could benefit from a more in-depth study, since the evidence in favor of FS on the basis of quality alone is not as strong as for FFNN.

With regard to model complexity, the networks evaluated are quite shallow: CNN and RNN models only have one hidden layer, and FFNN can have two at most. In spite of this, they have been able to produce at least partially good results (FFNN and RNN in dataset 104), and even results comparable to deeper networks (CNN against DBN) across all datasets. Incidentally, shallow networks appear to be more effective according to the literature on EEG-based BCI.

In turn, model complexity has a direct impact on optimization and training cost, be it for the amount of input features or for the properties of each network. It may also be true that some networks leave less room for tweaking, as for instance CNN probably needs to use the full 3, 600 features to reach its potential, and FFNN probably benefits the most from subsets of under 30 features. In any case, the vast differences in time and energy consumption among the three alternatives studied mean that available computing resources must be taken into account to strike a balance between quality and feasibility. CNN provides the best results on average but at the highest cost, which may not always be practical, while FFNN and RNN are cheaper but not as competitive in quality. Furthermore, the variability of intra-class cost-quality ratios is also a factor to keep in mind: there are often solutions with very similar accuracies but contrasting training costs, i.e., smaller networks can be competitive too. This could motivate introducing some mechanisms into optimization procedures to try to leverage this insight.

With cost-saving in mind, future work could take a more in-depth look at the differences between networks of the same type but varying sizes, generalize this analysis to different datasets, or tackle the tuning of neural network hyperparameters by using other techniques, such as Bayesian optimization to guide the search in a more informed (and thus, perhaps more efficient) way.

## Supporting information

**S1 Table. Hyperparameter description of the CNN models used in the comparisons.** (PDF)

**S2 Table. Hyperparameter description of the FFNN models used in the comparisons.**
(PDF)

**S3 Table. Hyperparameter description of the RNN models used in the comparisons.**
(PDF)

**S1 Data.**
(ZIP)

## Author Contributions

**Conceptualization:** Javier León, Juan José Escobar, Andrés Ortiz, Julio Ortega, Pedro Martín-Smith.

**Data curation:** Julio Ortega, John Q. Gan.

**Formal analysis:** Javier León.

**Funding acquisition:** Julio Ortega, Jesús González, Miguel Damas.

**Investigation:** Javier León.

**Methodology:** Javier León, Andrés Ortiz.

**Project administration:** Jesús González, Miguel Damas.

**Resources:** John Q. Gan.

**Software:** Javier León, Juan José Escobar.

**Supervision:** Julio Ortega, Jesús González, John Q. Gan, Miguel Damas.

**Validation:** Javier León, Andrés Ortiz.

**Visualization:** Javier León, Juan José Escobar.

**Writing – original draft:** Javier León, Andrés Ortiz, Julio Ortega, Pedro Martín-Smith, John Q. Gan.

**Writing – review & editing:** Javier León, Juan José Escobar, Andrés Ortiz, Julio Ortega, Pedro Martín-Smith, Miguel Damas.

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
