## [Decision Letter · Decision Letter 0]

3 Mar 2020

PONE-D-19-33145

A two-step approach to neural network hyperparameter optimization for EEG classification

PLOS ONE

Dear Mr. León,

Thank you for submitting your manuscript to PLOS ONE. After careful consideration, we feel that it has merit but does not fully meet PLOS ONE’s publication criteria as it currently stands. Therefore, we invite you to submit a revised version of the manuscript that addresses the points raised during the review process.

The reviewers require more details on the personal contribution, justification of the architecture of the CNN, more information on the GA parametrization, comparison of the methodology to existing approaches, extension to other data sets, enhancement of the state-of-the-art description.

We would appreciate receiving your revised manuscript by Apr 11 2020 11:59PM. To enhance the reproducibility of your results, we recommend that if applicable you deposit your laboratory protocols in protocols.io, where a protocol can be assigned its own identifier (DOI) such that it can be cited independently in the future. For instructions see: http://journals.plos.org/plosone/s/submission-guidelines#loc-laboratory-protocols

We look forward to receiving your revised manuscript.

Kind regards,

Ruxandra Stoean

Academic Editor

PLOS ONE

Journal Requirements:

3. Your ethics statement must appear in the Methods section of your manuscript. If your ethics statement is written in any section besides the Methods, please move it to the Methods section and delete it from any other section. Please also ensure that your ethics statement is included in your manuscript, as the ethics section of your online submission will not be published alongside your manuscript.

Reviewers' comments:

Reviewer's Responses to Questions

**Comments to the Author**

1. Is the manuscript technically sound, and do the data support the conclusions?

Reviewer #1: Yes

Reviewer #2: Partly

2. Has the statistical analysis been performed appropriately and rigorously? 

Reviewer #1: Yes

Reviewer #2: No

3. Have the authors made all data underlying the findings in their manuscript fully available?

Reviewer #1: Yes

Reviewer #2: No

4. Is the manuscript presented in an intelligible fashion and written in standard English?

Reviewer #1: Yes

Reviewer #2: Yes

5. Review Comments to the Author

Reviewer #1: This paper address the analysis of brain activity via Electroencephalography (EEG). EGG patterns are classified into three classes that represent imagined left and right hand movements and imagined feet movement. With this aim a two-step genetic approach to neural network hyperparameter optimization is proposed. The neural network developed in a genetic way are a Convolutional Neural Networks (CNNs), a Feed-Forward Neural Networks(FFNNs), and Recurrent Neural Networks (RNNs).

One remarkable characteristic of the dataset is that each patter contains 3600 features but on 178 patterns are available to train.

The paper is well structured and well written. The experimentation is correctly designed and the results are validated with appropriate statistical tools. In this way contents can be considered technically sound.

However the paper have certain drawbacks that I recommend to overcome:

1. In my opinion neither the title of the paper nor abstract properly reflect/represent the use of genetic algorithms or the use automatic tools to the development of neural networks. I think this aspect can be enhanced in both parts, mainly in the abstract.

2. The article lacks a literature review about the EGG classification research area because only some papers about mainly belonging to the authors themselves are cited. Authors have to insert this state of art review.

3. Authors do not include a justification and/or an enumeration of the contributions of their paper. I think that using the review of the previous item, authors can fulfill this requirement.

4. Authors sentence that use a two-step approach to neural network optimization, however it is not clear how this approach works. Is the best individual obtained from the first stage chosen as the initial individual of the second stage?. This methodology must be clarified as well as the initialization step of the three genetic algorithms used. A figure could enhance these explanations.

5. For FFNN a description for the available activation functions is made but I did not see in the paper the function used in the paper. A justification of this choice should be inserted.

6. Regarding the parameters of the GA:

o The number of generations for the structure optimization GA and for the learning optimization GA seems a bit low. Have you studied the fitness convergence of these GAs. Please, justify the choice.

o The topology to optimize for CNN is unusual and shallow. CNNs often have more than one layer, at least two, and a pooling layer. Can you justify this choice? Is there another examples of this topology in the literature.

o The only constrain number in the for the FFNN structure is 2 hidden layers. But what is the width of each layer.

o How the constrain structure numbers (250, 2, 60,…) are chosen? A priori, may seems there are not a fair competition.

7. The best individuals (widths, epochs, …) obtained for each type of NN are not shown. In my opinion it is very interesting showing these best individuals and analyzing the traits of them

Reviewer #2: The authors have extended their previous work and have used genetic algorithm (GA) in 2 stages to optimize 3 different network models (CNN, FFNN and RNN). The paper lacks novelty, is not clear and has not been evaluated properly. Following are my specific comments:

1. The paper lacks novelty as use of GA for optimization of network parameters is not new. Furthermore, the paper mostly presents the details of the 3 different models, which is already widely available and a detailed information on the actual work is missing from the paper. Hyper-parameter optimization can also be done using Bayesian optimization (for eg. [1]), how does GA compare with Bayesian optimization and what are the advantages of using GA?

2. The authors have used only 3 subject’s data to evaluate the performance of their method. It cannot be generalized that the proposed method is good by only evaluating it using data of 3 subjects. More evaluation and analysis is required using larger datasets such as the GigaDB dataset [2].

3. There is no comparison of the proposed method with the existing state-of-the-art-methods. Some of the recent state-of-the-art methods are listed below:

• S. Li and H. Feng, "EEG Signal Classification Method Based on Feature Priority Analysis and CNN," in 2019 International Conference on Communications, Information System and Computer Engineering (CISCE), 2019, pp. 403-406.

• S. Kumar, A. Sharma, and T. Tsunoda, "Brain wave classification using long short-term memory network based OPTICAL predictor," Scientific Reports, vol. 9, p. 9153, 2019.

• P. Gaur, R. B. Pachori, H. Wang, and G. Prasad, "A multi-class EEG-based BCI classification using multivariate empirical mode decomposition based filtering and Riemannian geometry," Expert Systems with Applications, vol. 95, pp. 201-211, 2018.

• S. Kumar, A. Sharma, and T. Tsunoda, "An improved discriminative filter bank selection approach for motor imagery EEG signal classification using mutual information," BMC Bioinformatics, vol. 18, p. 545, December 28 2017.

[1] S. Kumar, A. Sharma, and T. Tsunoda, "Brain wave classification using long short-term memory network based OPTICAL predictor," Scientific Reports, vol. 9, p. 9153, 2019.

[2] H. Cho, M. Ahn, S. Ahn, M. Kwon, and S. C. Jun, "EEG datasets for motor imagery brain–computer interface," GigaScience, vol. 6, pp. 1-8, 5th April 2017.

6. PLOS authors have the option to publish the peer review history of their article (what does this mean?). If published, this will include your full peer review and any attached files.

Reviewer #1: No

Reviewer #2: No

---

## [Author Response · Author response to Decision Letter 0]

7 May 2020

We have uploaded our response to the reviewers in the corresponding PDF file.

---

## [Editor Report · Decision Letter 1]

21 May 2020

Deep learning for EEG-based Motor Imagery classification: accuracy-cost trade-off

PONE-D-19-33145R1

Dear Dr. León,

We are pleased to inform you that your manuscript has been judged scientifically suitable for publication and will be formally accepted for publication once it complies with all outstanding technical requirements.

With kind regards,

Ruxandra Stoean

Academic Editor

PLOS ONE
---

## [Editor Report · Acceptance letter]

29 May 2020

PONE-D-19-33145R1 

Deep learning for EEG-based Motor Imagery classification: accuracy-cost trade-off 

Dear Dr. León:

I am pleased to inform you that your manuscript has been deemed suitable for publication in PLOS ONE. Congratulations! Your manuscript is now with our production department. 

With kind regards,

on behalf of

Dr. Ruxandra Stoean 

Academic Editor

PLOS ONE